# Superposition of many models into one

**Brian Cheung**
Redwood Center, BAIR
UC Berkeley
bcheung@berkeley.edu

**Alex Terekhov**
Redwood Center
UC Berkeley
aterekhov@berkeley.edu

**Yubei Chen**
Redwood Center, BAIR
UC Berkeley
yubeic@berkeley.edu

**Pulkit Agrawal**
BAIR
UC Berkeley
pulkitag@berkeley.edu

**Bruno Olshausen**
Redwood Center, BAIR
UC Berkeley
baolshausen@berkeley.edu

## Abstract

We present a method for storing multiple models within *a single set of parameters*. Models can coexist in *superposition* and still be retrieved individually. In experiments with neural networks, we show that a surprisingly large number of models can be effectively stored within a single parameter instance. Furthermore, each of these models can undergo thousands of training steps without significantly interfering with other models within the superposition. This approach may be viewed as the *online* complement of compression: rather than reducing the size of a network after training, we make use of the unrealized capacity of a network *during training*.

## 1  Introduction

While connectionist models have enjoyed a resurgence of interest in the artificial intelligence community, it is well known that deep neural networks are over-parameterized and a majority of the weights can be pruned *after* training [7, 20, 3, 8, 9, 1]. Such pruned neural networks achieve accuracies similar to the original network but with much fewer parameters. However, it has *not been possible* to exploit this redundancy to train a neural network with fewer parameters from scratch to achieve accuracies similar to its over-parameterized counterpart. In this work we show that it is possible to *partially* exploit the excess capacity present in neural network models *during training* by learning multiple tasks. Suppose that a neural network with $L$ parameters achieves desirable accuracy at a single task. We outline a method for *training a single neural network* with $L$ parameters to simultaneously perform $K$ different tasks and thereby effectively requiring $\approx \mathcal{O}\left(\frac{L}{K}\right)$ parameters per task.

While we learn a separate set of parameters $\left(W_k; \ k \in [1, K]\right)$ for each of the K tasks, these parameters are stored in *superposition* with each other, thus requiring approximately the same number of parameters as a model for a single task. The task-specific models can be accessed using task-specific "context" information $C_k$ that dynamically "routes" an input towards a specific model retrieved from this superposition. The model parameters $W$ can be therefore thought of as a "memory" and the context $C_k$ as "keys" that are used to access the specific parameters $W_k$ required for a task. Such an interpretation is inspired by Kanerva's work on hetero-associative memory [4].

Because the parameters for different tasks exist in super-position with each other and are constantly changing during training, it is possible that these individual parameters interfere with each other and thereby result in loss in performance on individual tasks. We show that under mild assumptions of the input data being intrinsically low-dimensional relative to its ambient space (e.g. natural images lie on a much lower dimensional subspace as compared to their representation of individual pixels

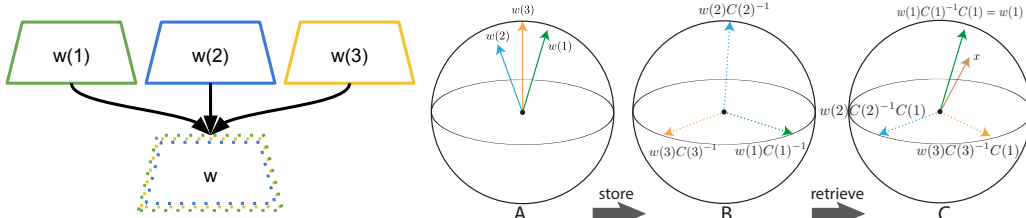

Figure 1: Left: Parameters for different models $w(1)$, $w(2)$ and $w(3)$ for different tasks are stored in superposition with each other in $w$. Right: To prevent interference between (**A**) similar set of parameter vectors $w(s), s \in \{1, 2, 3\}$, we **B (store)** these parameters after rotating the weights into nearly orthogonal parts of the space using task dependent context information $(C^{-1}(s))$. An appropriate choice of $C(s)$ ensures that we can **C (retrieve)** $\hat{w}(k)$ by operation $wC(k)$ in a manner that $w(s)$, for $s \neq k$ will remain nearly orthogonal, reducing interference during learning.

with RGB values), it is possible to choose *context* that minimizes such interference. The proposed method has wide ranging applications such as training a neural networks in memory constrained environments, online learning of multiple tasks and over-coming catastrophic forgetting.

**Application to Catastrophic Forgetting:** Online learning and sequential training of multiple tasks has traditionally posed a challenge for neural networks. If the distribution of inputs (e.g. changes in appearance from day to night) or the distribution output labels changes over time (e.g. changes in the task) then training on the most recent data leads to poor performance on data encountered earlier. This problem is known as catastrophic forgetting [12, 15, 2]. One way to deal with this issue is to maintain a memory of all the data and train using batches that are constructed by uniformly and randomly sampling data from this memory (replay buffers [14]). However in memory constrained settings this solution is not viable. Some works train a separate network (or sub-parts of network) for separate task [17, 19, 11]. The other strategy is to selectively update weights that do not play a critical role on previous tasks using variety of criterion such as: Fisher information between tasks [5], learning an attention mask to decide which weights to change [10, 18] and other criterion [22]. However, these methods prevent re-use of weights in the future and therefore intrinsically limit the capacity of the network to learn future tasks and increase computational cost. Furthermore, for every new task, one additional variable per weight parameter indicating whether this weight can be modified in the future or not (i.e. $L$ new parameters per task) needs to be stored.

We propose a radically different way of using the same set of parameters in a neural network to perform multiple tasks. We store the weights for different tasks in superposition with each other and do not explicitly constrain how any specific weight parameter changes within the superposition. Furthermore, we need to store substantially less additional variables per new task (1 additional variable per task for one variant of our method; Section 2.1). We demonstrate the efficacy of our approach of learning via *parameter superposition* on two separate online image-classification settings: (a) time-varying input data distribution and (b) time-varying output label distribution. With parameter superposition, it is possible to overcome catastrophic forgetting on the permuting MNIST [5] task, continuously changing input distribution on rotating MNIST and fashion MNIST tasks and when the output labels are changing on the incremental CIFAR dataset [16].

## 2 Parameter Superposition

The intuition behind *Parameter Superposition (PSP)* as a method to store many models simultaneously into one set of parameters stems from analyzing the fundamental operation performed in all neural networks – multiplying the inputs $(x \in \Re^N)$ by a weight matrix $(W \in \Re^{M \times N})$ to compute features $(y = Wx)$. Over-parameterization of a network essentially implies that only a small sub-space spanned by the rows of $W$ in $\Re^N$ are relevant for the task.

Let $W_1, W_2, ..., W_K$ be the set of parameters required for each of the $K$ tasks. If only a small subspace in $\Re^N$ is required by each $W_k$, it should be possible to transform each $W_k$ using a task-specific linear transformation $C_k^{-1}$ (that we call as *context*), such that rows of each $W_k C_k^{-1}$ occupy mutually orthogonal subspace in $\Re^N$ (see Figure 1). Because each $W_k C_k^{-1}$ occupies a different

|            | # parameters   | +1 model |
|------------|----------------|----------|
| Standard   | $MN$           | $MN$     |
| Rotational | $M(N+M)$       | $M^2$    |
| Binary     | $M(N+1)$       | $M$      |
| Complex    | $2M(N+0.5)$    | $M$      |
| OnePower   | $2M(N+0.5)$    | $1$      |

Table 1: Parameter count for superposition of a linear transformation of size $L = M \times N$. '+1 model' refers to the number of additional parameters required to add a new model.

subspace, these parameters can be summed together without interfering when stored in superposition:

$$W = \sum_{i=1}^{K} W_i C_i^{-1} \tag{1}$$

This is similar to the superposition principle in *fourier analysis* where a signal is represented as a superposition of sinusoids. Each sinusoid can be considered as the "context". The parameters for an individual task can be retrieved using the context $C_k$ and let them be referred by $\hat{W}_k$:

$$\hat{W}_k = WC_k = \sum_{i=1}^{K} W_i \big( C_i^{-1} C_k \big) \tag{2}$$

Because the weights are stored in superposition, the retrieved weights ($\hat{W}_k$) are likely to be a noisy estimate of $W_k$. Noisy retrieval will not affect the overall performance if $\hat{W}_k x = W_k x + \epsilon$, where $\epsilon$ stays small. A detailed analysis of $\epsilon$ for some choices of context vectors described in Section 2.1 can be found in the Appendix A.

In the special case of $C_k^{-1} = C_k^T$, each $C_k$ would be an orthogonal matrix representing a rotation. As matrix multiplication is associative, $y_k = (WC_k)x$ can be rewritten as $y_k = W(C_k x)$. The PSP model for computing outputs for the $k^{th}$ task is therefore,

$$y_k = W\big(C_k x\big) \tag{3}$$

In this form PSP can be thought of as learning a single set of parameters $W$ for multiple tasks, after the rotating the inputs ($x$) into orthogonal sub-spaces of $\Re^N$. It is possible to construct such orthogonal rotations of the input when $x$ itself is over-parameterized (i.e. it lies on a low-dimensional manifold). The assumption that $x$ occupies a low-dimensional manifold is a mild one and it is well known that natural signals such as images and speech do indeed have this property.

## 2.1 Choosing the Context

**Rotational Superposition** The most general way to choose the context is to sample rotations uniformly from orthogonal group $O(M)$ (Haar distribution)[1]. We refer to this formulation as *rotational superposition*. In this case, $C_k \in \Re^{M \times M}$ and therefore training for a new task would require $M^2$ more parameters. Thus, training for $K$ tasks would require $MN + (K-1)M^2$ parameters. In many scenarios $M \sim N$ and therefore learning by such a mechanism would require approximately as many parameters as training a separate neural network for each task. Therefore, the rotational superposition in its most general is not memory efficient.

It is possible to reduce the memory requirements of rotational superposition by restricting the context to a subset of the orthogonal group, e.g. random permutation matrices, block diagonal matrices or diagonal matrices. In the special case, we choose $C_k = diag(c_k)$ to be a diagonal matrix with the diagonal entries given by the vector $c_k$. With such a choice, only $M$ additional parameters are required per task (see Table 1). In case of a diagonal context, PSP in equation 3 reduces to an element-wise multiplication (symbol $\odot$) between $c_k, x$ and can be written as:

$$y = W(c(k) \odot x) \tag{4}$$

There are many choices of $c_k$ that lead to construction of orthogonal matrices:

**Complex Superposition**    In Equation 4, we can chose $c_k$ to be a vector of complex numbers, where each component $c_k^j$ is given by,

$$c_k^j = e^{i\phi_j(k)} \tag{5}$$

Each of the $c_k^j$ lies on the complex unit circle. The phase $\phi_j(k) \in [-\pi, \pi]$ for all $j$ is sampled with uniform probability density $p(\phi) = \frac{1}{2\pi}$. It can be seen that such a choice of $c_k$ results in a diagonal orthogonal matrix.

**Powers of a single context**    The memory footprint of complex superposition can be reduced to a *single parameter per task*, by choosing context vectors that are integer powers of one context vector:

$$c_k^j = e^{i\phi_j k} \tag{6}$$

**Binary Superposition**    Constraining the phase to two possible values $\phi_j(k) \in \{0, \pi\}$ is a special case of complex superposition. The context vectors become $c(k)_j \in \{-1, 1\}$. We refer to this formulation as *binary superposition*. The low-precision of the context vectors in this form of superposition has both computational and memory advantages. Furthermore, binary superposition is directly compatible with both real-valued and low-precision linear transformations.

## 3   Neural Network Superposition

We can extend these formulations to entire neural network models by applying superposition (Equation 3) to the linear transformation of all layers $l$ of a neural network:

$$x^{(l+1)} = g(W^{(l)}(c(k)^{(l)} \odot x^{(l)})) \tag{7}$$

where $g()$ is a non-linearity (e.g. ReLU).

**Extension to Convolutional Networks**    For neural networks applied to vision tasks, convolution is currently the dominant operation in a majority of layers. Since the dimensionality of convolution parameters is usually much smaller than the input image, it makes more sense computationally to apply context to the weights rather than the input. By associativity of multiplication, we are able reduce computation by applying a context tensor $c(k) \in \mathbb{C}^{M \times H_w \times W_w}$ to the convolution kernel $w \in \mathbb{C}^{N \times M \times H_w \times W_w}$ instead of the input image $x \in \mathbb{C}^{M \times H_x \times W_x}$:

$$y_n = (w_n \odot c(k)) * x \tag{8}$$

where $*$ is the convolution operator, $M$ is the input channel dimension, $N$ is the output channel dimension.

## 4   Experiments

There are two distinct ways in which the data distribution can change over time: (a) change in the input data distribution and (b) change in the output labels over time. Neural networks are high-capacity learners and can even learn tasks with random labelling [23]. Despite shifts in data distribution, if data from different tasks are pooled together and uniformly sampled to construct training batches, neural networks are expected to perform well. However, a practical scenario of interest is when it is not possible to access all the data at once, and online training is necessary. In such scenarios, training the neural network on the most recent task leads to loss in performance on earlier tasks. This problem is known as catastrophic forgetting – learning on one task interferes with performance on another task. We evaluate the performance the proposed PSP method on mitigating the interference in learning due to changes in input and output distributions.

### 4.1   Input Interference

A common scenario in online learning is when the input data distribution changes over time (e.g. visual input from day to night). Permuting MNIST dataset [2], is a variant of the MNIST dataset [7] where the image pixels are permuted randomly to create new tasks over time. Each permutation of pixels corresponds to a new task. The output labels are left unchanged. Permuting MNIST has been

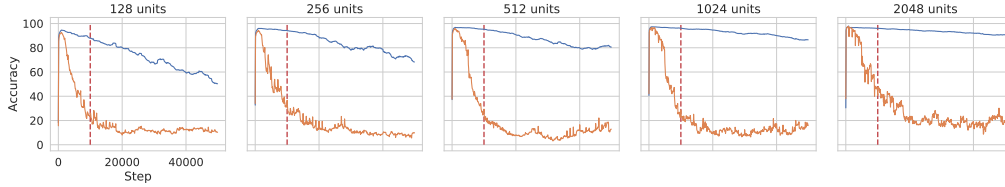

Figure 2: Comparing the accuracy of the binary superposition model (blue) with the baseline model (orange) for varying number of units in fully connected networks with differing number of units (128 to 2048) on the permuting MNIST challenge. On this challenge, the inputs are permuted after every 1000 iterations, and each permutation corresponds to a new task. 50K iterations therefore correspond to 50 different tasks presented in sequence. Dotted red line indicates completion of 10 tasks. It is to be expected that larger networks can fit more data and be more robust to catastrophic forgetting. While indeed this is true and the baseline model does better with more units, the PSP model is far superior and the effect of catastrophic forgetting is negligible in larger networks.

used by many previous works [2, 22, 17, 5] to study the problem of catastrophic forgetting. In our setup, a new task is created after every 1000 mini-batches (steps) of training by permuting the image pixels. To adapt to the new task, all layers of the neural network are finetuned. Figure 2 shows that a standard neural network suffers from catastrophic forgetting and the performance on the first task degrades after training on newer tasks.

Separate context parameters are chosen for every task. Each choice of context can be thought of as creating a new model within the same neural network that can be used to learn a new task. In case of binary superposition (Section 2.1), a random binary vector is chosen for each task, for complex superposition (Section 2.1), a random complex number (constant magnitude, random phase) is chosen and for rotation superposition (Section 2.1) a random orthogonal matrix is chosen. Note that use of task identity information to overcome catastrophic forgetting is not special to our method, but has been used by all previous methods [22, 5, 17]. We investigated the efficacy of PSP in mitigating forgetting with changes in network size and the methods of superposition.

### 4.1.1 Effect of network size on catastrophic forgetting

Bigger networks have more parameters and can thus be expected to be more robust to catastrophic forgetting as they can fit to larger amounts of data. We trained fully-connected networks with two hidden layers on fifty permuting MNIST tasks presented sequentially. The size of hidden layers was varied from 128 to 2048 units. Results in Figure 2 show marginal improvements in performance of the *standard* neural network with its size. The PSP method with binary superposition (*pspBinary*) is significantly more robust to catastrophic forgetting as compared to the standard baseline. Because higher number of parameters create space to pack a larger number of models in super-position, the performance of *pspBinary* also improves with network size and with hidden layer of size 2048, the performance on the initial task is virtually unchanged even after training for 49 other tasks with very different input data distribution.

### 4.1.2 Effect of types of superposition on catastrophic forgetting

Different methods of storing models in superposition use a different number of additional parameters per task. While *pspBinary* and *pspComplex* require $M$ (where $M$ is the size of the input to each layer for a fully-connected network) additional parameters; *pspRotation* requires $M^2$ additional parameters (see Table 1). Larger number of parameters implies that a set of more general orthogonal transformation that span larger number of rotations can be constructed. More rotations means that inputs can be rotated in more ways and thereby more models can be packed with the same number of parameters. Results shown in Figure 3 left confirm this intuition for networks of 256 units. Better performance of *pspComplex* as compared to *pspBinary* is not surprising because binary superposition is a special case of complex superposition (see section 2.1). In the appendix, we show these differences become negligible for larger networks.

While the performance of *pspRotation* is the best among all superposition methods, this method is impractical because it amounts to adding the same number of additional parameters as required

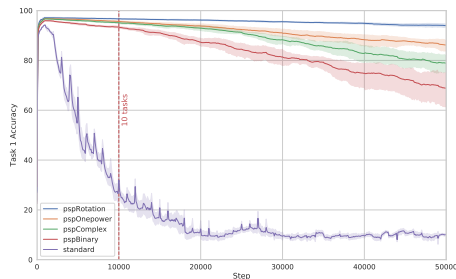

| | Avg. Accuracy (%) |
|---|---|
| EWC [5]* | 97.0 |
| SI [22]* | 97.2 |
| Standard | 61.8 |
| Binary | 97.6 |
| Complex | 97.4 |
| OnePower | 97.2 |

Figure 3: Left: Comparing the accuracy of various methods for PSP on the first task of the permuting MNIST challenge over training steps. After every 1000 steps, the input pixels are randomly permuted (i.e. new task) and therefore training on this newer task can lead to loss in performance on the initial task due to catastrophic forgetting. The PSP method is robust to catastrophic forgetting with pspRotation performing slightly better than pspComplex which in turn is better than pspBinary. This is expected as the number of additional parameters required per task in pspRotation > pspComplex > pspBinary (see Table 1). Right: The average accuracy over the last 10 tasks on the permuting MNIST challenge shows that the proposed PSP method outperforms previously published methods. *results from Figure 4 in Zenke et al. [22]

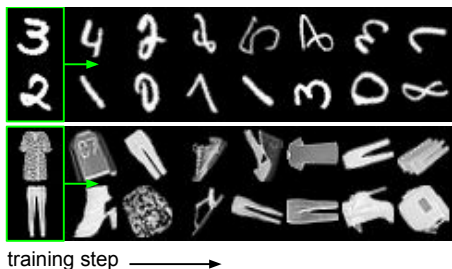

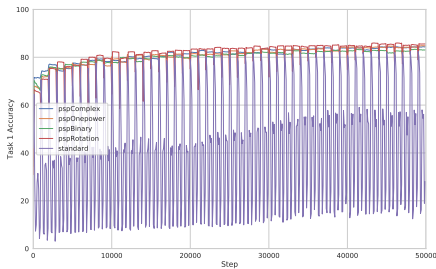

(a) Rotating (MNIST and fashionMNIST) datasets.  (b) Accuracy on fashionMNIST.

Figure 4: (a) Samples of rotating-MNIST (top) and rotating-FashionMNIST (bottom) datasets. To model a continuously and smoothly changing data stream, at every training step (i.e. mini-batch shown by green box), the images are rotated by a small counter-clockwise rotation. Images rotate by 360 degrees over 1000 steps. (b) Test accuracy for 0 degrees rotation as a function of number of training steps. A regular neural networks suffers from catastrophic forgetting. High accuracy is achieved after training on $0^o$ rotation and then the performance degrades. The proposed PSP method is robust to slow changes in data distribution when provided with the appropriate context.

for training a separate network for each task. *pspComplex* requires extension of neural networks to complex numbers and *pspBinary* is easiest to implement. To demonstrate the efficacy of our method, in the remainder of the paper we present most results with *pspBinary* with an understanding that *pspComplex* can further improve performance.

**Comparison to previous methods:** Table in Figure 3 compares the performance of our method with two previous methods: EWC [5] and SI [22]. Following the metric used in these works, we report the average accuracy on the last ten permuted MNIST tasks after the end of training on 50 tasks. PSP outperforms previous methods.

### 4.1.3 Continuous Domain Shift

While permuting MNIST has been used by previous work for addressing catastrophic forgetting, the big and discrete shift in input distribution between tasks is somewhat artificial. In the natural world, distribution usually shifts slowly – for example day gradually comes night and summer gradually becomes winter. To simulate real-world like continuous domain shift, we propose *rotating-MNIST*

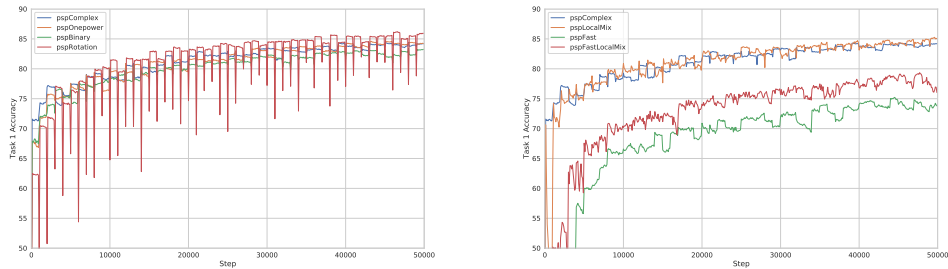

Figure 5: Left: Closer comparison of each form of parameter superposition on the rotating-FashionMNIST task at angle $0°$. Right: Different context selection functions on the rotating-FashionMNIST task at angle $0°$.

and *rotating-FashionMNIST* that are variants of the original MNIST and FashionMNIST [21] datasets. At every time step, the input images are rotated in-plane by a small amount in counter-clockwise direction. A rotation of $360^o$ is completed after 1000 steps and the input distribution becomes similar to the initial distribution. Every 1000 steps one complete cycle of rotation is completed. Sample images from the rotating datasets are shown in Figure 4a.

It is to be expected that very small rotations will not lead to interference in learning. Therefore, instead of choosing a separate context for every time step, we change the context after every 100 steps. The 10 different context vectors used in the first cycle (1000 steps) and are re-used in subsequent cycles of rotations. Figure 4b plots accuracy on a test data set of fashion MNIST with $0^o$ rotation with time. The oscillations in performance of the *standard* network correspond to 1000 training steps, which is the time required to complete one cycle of rotation. As the rotation of input images shifts away from $0^o$, due to catastrophic forgetting, the performance worsens and it improves as the cycle nears completion. The proposed PSP models are robust to changes in input distribution and closely follow the same trends as on permuting MNIST. These results show that efficacy of PSP approach is not specific to a particular dataset or the nature of change in the input distribution.

**Choosing context parameters:** Instead of using task identity to choose the context, it would be ideal if the context could be automatically chosen without this information. While a general solution to this problem is beyond the scope of this paper, we investigate the effect of using looser information about task identity on catastrophic forgetting. For this we constructed, *pspFast* a variant of *pspComplex* where the context is randomly changed at every time step for 1000 steps corresponding to one cycle of rotations. In the next cycle these contexts are re-used. In this scenario, instead of using detailed information about the task identity only coarse information about when the set of tasks repeat is used. Absence of task identity requires storage of 1000 models in superposition, which is 100x times the number of models stored in previous scenarios. Figure 5 right shows that while *pspFast* is better than *standard* model, it is worse in performance when more detailed task identity information was used. Potential reasons for worse performance are that each model in *pspFast* is trained with lesser amount of data (same data, but 100x models) and increased interference between models stored.

Another area of investigation is the scenario when detailed task information is not available, but some properties about changes in data distribution are known. For example, in the rotating fashion MNIST task it is known that the distribution is changing slowly. In contrast to existing methods, one of the strengths of the PSP method is that it is possible to incorporate such knowledge in constructing context vectors. To demonstrate this, we constructed *pspFastLocalMix*, a variant of *pspFast*, where at every step we define a context vector as a mixture of the phases of adjacent timepoints. Figure 5 shows that *pspFastLocalMix* leads to better performance than *pspFast*. This provides evidence that it is indeed possible to incorporate coarse information about non-stationarity of input distribution.

## 4.2 Output Interference

Learning in neural networks can be adversely affected by changes in the output (e.g. label) distribution of the training data. For example, this occurs when transitioning from one classification task to another. The incremental CIFAR (iCIFAR) dataset [16, 22] (see Figure 6a) is a variant of the CIFAR

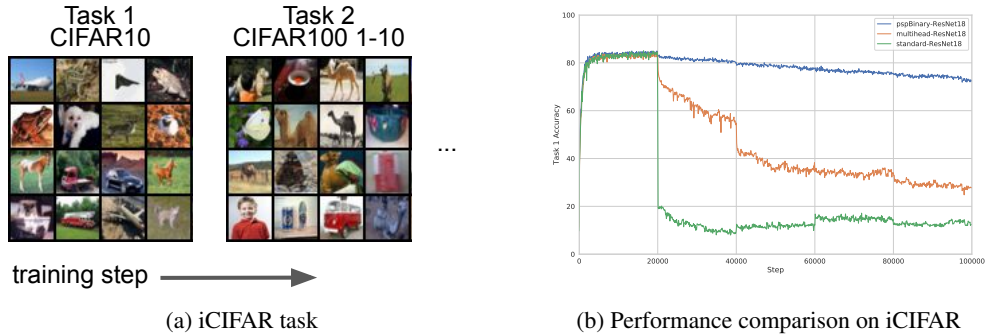

(a) iCIFAR task            (b) Performance comparison on iCIFAR

Figure 6: (a) Samples from the iCIFAR dataset. (b) Accuracy of ResNet-18 model on CIFAR-10 test dataset after training for 20K steps first on CIFAR-10 dataset, and then sequentially finetuning on four disjoint set of 10 classes from CIFAR-100 for 20K iterations each. The baseline *standard* and *multihead* model are critically affected by changes in output labels, whereas the PSP model with binary superposition is virtually unaffected. These result shows the efficacy of PSP in dealing with catastrophic forgetting and easy scaling to state-of-the-art neural networks.

dataset [6] where the first task is the standard CIFAR-10 dataset and subsequent tasks are formed by taking disjoint subsets of 10 classes from the CIFAR-100 dataset.

To show that our PSP method can be used with state-of-the-art neural networks, we used ResNet-18 to first train on CIFAR-10 dataset for 20K steps. Next, we trained the network on 20K steps on four subsequent and disjoint sets of 10 classes chosen from the CIFAR-100 dataset. We report the performance on the test set of CIFAR-10 dataset. Unsurprisingly, the standard ResNet-18 suffers a big loss in performance on the CIFAR-10 after training on classes from CIFAR-100 (see *standard-ResNet18* in Figure 6b). This forms a rather weak baseline, because the output targets also changes for each task and thus reading out predictions for CIFAR-10 after training on other tasks is expected to be at chance performance. A much stronger baseline is when a new output layer is trained for every task (but the rest of the network is re-used). This is because, it might be expected that for these different tasks the same features might be adept but a different output readout is required. Performance of a network trained in this fashion, *multihead-ResNet18*, in Figure 6b is significantly better than *standard-ResNet18*.

To demonstrate the robustness of our approach, we train ResNet-18 with binary superposition on iCIFAR using only *a single output layer* and avoiding the need for a network with multiple output heads in the process. The PSP network suffers surprisingly little degradation in accuracy despite significant output interference.

## 5    Discussion

We have presented a fundamentally different way of diminishing catastrophic forgetting via storing multiple parameters for multiple tasks in the same neural network via superposition. Our framework treats neural network parameters as memory, from which task-specific model is retrieved using a context vector that depends on the task-identity. Our method works with both fully-connected nets and convolutional nets. It can be easily scaled to state-of-the-art neural networks like ResNet. Our method is robust to catastrophic forgetting caused due to both input and output interference and outperforms existing methods. An added advantage of our framework is that it can easily incorporate coarse information about changes in task distribution and does not completely rely on task identity (see Section 4.1.3). Finally, we proposed the rotating MNIST and rotating fashion MNIST tasks to mimic slowly changing task distribution that is reflective of the real world.

While in this work we have demonstrated the utility of PSP method, a thorough analysis of how many different models can be stored in superposition with each other will be very useful. This answer is likely to depend on the neural network architecture and the specific family of tasks. Another very interesting avenue of investigation is to automatically and dynamically determine context

vector instead of relying on task-specific information. One fruitful direction is to make the context differentiable instead of using a fixed context.

## Footnotes

[1]we use `scipy.stats.ortho_group`.

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
