[Supplementary Material · psp_camera_ready_supp.pdf]

# Supplementary Material

We establish properties of the destructive interference which make it possible to recover a linear transformation from the superposition. Appendix A provides intuition that after offline training, the models can be stored in superposition and retrieved with small noise. Appendix B shows that the models in superposition can be trained online. Appendix D describes how complex vectors can be generated compositionally.

Properties of this recovery process are more clearly illustrated by substituting Equation 1 into Equation 3. By unpacking the inner product operation, Equation 3 can be rewritten as the sum of two terms which is shown in Equation 9.

$$y_i = \sum_j \sum_s W(s)_{ij} c(s)_j^{-1} c(k)_j x_j$$
$$= \sum_j W(k)_{ij} x_j + \sum_j \sum_{s \neq k} W(s)_{ij} c(s)_j^{-1} c(k)_j x_j$$

which is written more concisely in matrix notation as:

$$y = W(k)x + \epsilon \tag{9}$$
$$\epsilon = \sum_{s \neq k} W(s)(c(s)^{-1} \odot c(k) \odot x)$$

The first term, $W(k)x$, is the recovered linear transformation and the second term, $\epsilon$, is a residual. For particular formulations of the set of context vectors $c(S)$, $\epsilon$ is a summation of terms which interfere destructively. For an analysis of the interference, please see the Appendix A.

## A  Analysis of retrieval noise

In this section, we use Propositions 1 and 2 to provide the intuition that we can superimpose individual models after training and the interference should stay small. Assume $w$ and $x$ are fixed vectors and $c$ is a random context vector, each element of which has a unit amplitude and uniformly distributed phase.

### A.1  Proposition 1: superposition bias analysis

**Proposition 1.** *$\epsilon$ in expectation is unbiased, $E_s[\epsilon] \to 0$.*

Proposition 1 states that, in expectation, other models within the superposition will not introduce a bias to the recovered linear transformation.

*Proof.* We consider three cases: real value network with binary context vectors, complex value network with complex context vectors, and real value network with orthogonal matrix context. For each case we show that if the context vectors / matrices have uniform distribution on the domain of their definition the expectation of the scalar product with the context-effected input vector is zero.

**Real-valued network with binary context vectors.**  Assuming a fixed weights vector $w$ for a given neuron, a fixed pre-context input $x$, and a random binary context vector $b$ with i.i.d. components

$$E\left[(w \odot b, x)\right] = E\left[\sum w_i b_i x_i\right]$$
$$= \sum w_i x_i E\left[b_i\right]$$
$$= 0$$

because $E\left[b_i\right] = 0$.

**Complex-valued network with complex context vectors.** Again, we assume a fixed weights vector $w$ for a given neuron, a fixed pre-context input $x$, and a random complex context vector $c$ with i.i.d. components, such that $\|c_i\| = 1$ for every $i$ and the phase of $c_i$ has a uniform distribution on a circle. Then

$$\mathrm{E}\left[(w \odot c, x)\right] = \mathrm{E}\left[\sum w_i^* c_i^* x_i\right]$$
$$= \sum w_i^* x_i \mathrm{E}\left[c_i^*\right]$$
$$= 0$$

where $w_i^*$ is a conjugate of $w_i$. Here $\mathrm{E}\left[c_i^*\right] = 0$ because $c_i$ has a uniform distribution on a circle.

**Real-valued network with real-valued rotational context matrices.** Let $w \in \mathbb{R}^M$ be a vector and the context vector $C_k$ be a random orthogonal matrix drawn from the $O(M)$ Haar distribution [13]. Then $C_k w$ for fixed $w$ defines a uniform distribution on sphere $S_r^{M-1}$, with radius $r = \|w\|$. Due to the symmetry,

$$\mathrm{E}\left[\langle C_k w, x \rangle\right] = 0.$$

$\square$

## A.2 Proposition 2: Variance induced by context vectors

**Proposition 2.** *For $x$, $w \in \mathbb{C}^M$ , when we bind a random $c \in \mathbb{C}^M$ with $w$, $\frac{\mathrm{Var}\left[\langle c \odot w, x \rangle\right]}{\|w\|^2 \|x\|^2} \approx \frac{1}{M}$ under mild conditions. For $x$, $w \in \mathbb{R}^M$, let $C_k$ be a random orthogonal matrix s.t. $C_k w$ has a random direction. Then $\frac{\mathrm{Var}\left(\langle C_k w, x \rangle\right)}{\|w\|^2 \|x\|^2} \approx \frac{1}{M}$. In both cases, let $|\langle w, x \rangle| = \|w\|\|x\|\eta$. If $\eta$ is large, then $|\langle c \odot w, x \rangle|$ and $|\langle C_k w, x \rangle|$ will be relatively small compared to $|\langle w, x \rangle|$.*

If we assume that $\|w(k)\|$'s are equally large and denote it by $\gamma$, then $\epsilon \propto \frac{K-1}{M}|\langle w, x \rangle|^2$. When $\frac{K-1}{M}$ is small, the residual introduced by other superimposed models will stay small. Binding with the random keys roughly attenuates each model's interference by a factor proportional to $\frac{1}{\sqrt{M}}$.

*Proof.* Similarly to Proposition 1, we give an estimate of the variance for each individual case: a real-valued network with binary context vectors, a complex-valued network with complex context vectors, and a real-valued network with rotational context matrices. All the assumptions are the same as in Proposition 1.

**Real-valued network with binary context vectors.**

$$\mathrm{E}\left[|(w \odot b, x)|^2\right] = \mathrm{E}\left[\left(\sum w_i b_i x_i\right)\left(\sum w_j b_j x_j\right)\right]$$
$$= \mathrm{E}\left[\sum_{i,j} w_i b_i x_i w_j b_j x_j\right]$$
$$= \sum_{i,j} w_i x_i w_j x_j \mathrm{E}\left[b_i b_j\right]$$
$$= \sum_i w_i^2 x_i^2 \mathrm{E}\left[b_i^2\right]$$
$$= \sum_i |w_i|^2 |x_i|^2$$
$$= \|w \odot x\|^2.$$

Here we made use of the facts that $b_i$ and $b_j$ are independent variables with zero mean, and that $b_i^2 = 1$.

Note that

$$|\langle w, x \rangle| = \|w\|\|x\|\eta.$$

If $\eta \gg 0$, then $|\langle w \odot b, x \rangle|$ will be relatively small compared to $|\langle w, x \rangle|$. Indeed, we can assume that each term $w_i^* x_i$ has a comparably small contribution to the inner product (e.g. when using dropout) and

$$\frac{|w_i|}{\|w\|} \frac{|x_i|}{\|x\|} \approx \frac{\eta}{M}.$$

Then

$$\mathrm{E}\left[|(w \odot b, x)|^2\right] \approx \frac{\eta^2}{M} \|w\|^2 \|x\|^2$$

**Complex-valued network with complex context vectors.**

$$\begin{aligned}
\mathrm{E}\left[|(c \odot w, x)|^2\right] &= \mathrm{E}\left[\left(\sum w_i x_i^* c_i\right)\left(\sum w_j^* x_j c_j^*\right)\right] \\
&= \mathrm{E}\left[\sum_{i,j} w_i x_i^* c_i w_j^* x_j c_j^*\right] \\
&= \sum_{i,j} w_i x_i^* w_j^* x_j \mathrm{E}\left[c_i c_j^*\right] \\
&= \sum_i w_i x_i^* w_i^* x_i \mathrm{E}\left[c_i c_i^*\right] \\
&= \sum_i |w_i|^2 |x_i|^2 \\
&= \|w \odot x\|^2
\end{aligned}$$

Here we make use of the fact that $\mathrm{E}\left[c_i^* c_j\right] = \mathrm{E}[c_i]\mathrm{E}\left[c_j^*\right] = 0$ for $i \neq j$, which in turn follows from the fact that $c_i$ and $c_j$ are independent variables.

Let's assume each term $w_i^* x_i$ has a comparable small contribution to the inner product (e.g. when using dropout). Then it is reasonable to assume that $|w_i||x_i| \approx \frac{\eta}{M}\|w\|\|x\|$, where $M$ is the dimension of $x$. Then

$$\mathrm{E}\left[|(w, x \odot c)|^2\right] \approx \frac{\eta}{M}\|w\|^2 \|x\|^2$$

**Real-valued network with real-valued rotational context matrices.** We show the case in high dimensional real vector space for a random rotated vector. Let $w \in \mathbb{R}^M$ again be a vector and $C_k$ be a random matrix drawn from the $O(M)$ Haar distribution. Then $C_k w$ for fixed $w$ defines a uniform distribution on sphere $S_r^{M-1}$, with radius $r = \|w\|$.

Consider a random vector $w'$, whose components are drawn i.i.d. from $N(0, 1)$. Then one can establish a correspondence between $C_k w$ and $w'$:

$$C_k w = \|w\| \frac{w'}{\|w'\|}.$$

thus $C_k w$ is $w$ under a random rotation in $\mathbb{R}^M$. Let $w' = (w'_1, \ldots, w'_M)$ be a random vector where $w'_i$ are i.i.d. normal random variables $N(0, 1)$. $C_k w \sim \|w\| \frac{w'}{\|w'\|}$, $\langle w', x \rangle = \sum_{i=1}^M x_i w'_i$ and $\langle C_k w, x \rangle = \frac{\|w\|}{\|w'\|}\langle w', x \rangle$. Then we have:

$$\begin{aligned}
\mathrm{Var}\left(\langle C_k w, x \rangle\right) &= \mathrm{E}\left[\langle C_k\, w, x \rangle^2\right] \\
&= \mathrm{E}\left[\frac{\|w\|^2}{\|w'\|^2}\langle w', x \rangle^2\right] \\
&= \|w\|^2 \|x\|^2 \mathrm{E}\left[\langle \frac{w'}{\|w'\|}, \frac{x}{\|x\|}\rangle^2\right].
\end{aligned}$$

We further show that $\mathrm{E}\left[\langle \frac{w'}{\|w'\|}, \frac{x}{\|x\|}\rangle^2\right] = \frac{1}{M}$ and hence

$$\mathrm{Var}\left(\langle C_k \cdot w, x\rangle\right) = \frac{1}{M}\|w\|^2\|x\|^2.$$

Due to the symmetry,

$$\mathrm{E}\left[\langle \frac{w'}{\|w'\|}, \frac{x}{\|x\|}\rangle^2\right] = \mathrm{E}\left[\langle \frac{w'}{\|w'\|}, (1, 0, \ldots, 0)\rangle^2\right]$$
$$= \mathrm{E}\left[(\frac{w'_1}{\|w'\|})^2\right]$$

Let $y_i = \frac{w'_i}{\|w'\|}$, $\mathrm{E}\left[y_i^2\right] = \gamma$. Then

$$\gamma = \mathrm{E}\left[y_1^2\right] = \mathrm{E}\left[1 - \sum_{i=2}^{M} y_i^2\right]$$
$$= 1 - (M-1)\gamma$$

So $M\gamma = 1$ and $\gamma = \frac{1}{M}$. Thus $\mathrm{E}\left[(\frac{w'_1}{\|w'\|})^2\right] = \frac{1}{M}$.

Let $|\langle w, x\rangle| = \|w\|\|x\|\eta$, if we consider the case $\eta$ is large, we have $\mathrm{std}\left(\langle C_k w, x\rangle\right) \propto \frac{1}{\sqrt{M}}|\langle w, x\rangle|$.
□

## B   Online learning with unitary transformations

In this section, we describe having individual models in superposition during training.

**Proposition 3.** *Denote the cost function of the network with PSP as $J_{PSP}$ used in context $k$, and the cost function of the $k^{th}$ network without as $J_k$.*

1. *For the complex context vector case, $\frac{\partial}{\partial W} J_{PSP} \approx \left(\frac{\partial}{\partial W(k)} J_k\right) \odot c(k)$, where $c(k)$ is the context vector used in $J_{PSP}$ and $W(k)$ are the weights of the $J_k$ network.*

2. *For the general rotation case in real vector space, $\frac{\partial}{\partial W} J_{PSP} \approx (\frac{\partial}{\partial W(k)} J_k)C_k$, where $C_k$ is the context matrix used in $J_{PSP}$ and $W(k)$ are the weights of the $J_k$ network.*

Proposition 3 shows parameter updates of an individual model in superposition is approximately equal to updates of that model trained outside of superposition. The gradient of parameter superposition creates a *superposition of gradients* with analogous destructive interference properties to Equation 3. Therefore, memory operations in parameter superposition can be applied in an online fashion.

*Proof.* Here we show that training a model which is in superposition with other models using gradient descent yields almost the same parameter update as training this model independently (without superposition). For example imagine two networks with parameters $w_1$ and $w_2$ combined into one superposition network using context vectors $c_1$ and $c_2$, such that the parameters of the PSP network $w = w_1 \odot c_1 + w_2 \odot c_2$. Then, what we show below is that training the PSP network with the context vector $c_1$ results in nearly the same change of parameters $w$ as training the network $w_1$ independently and then combining it with $w_2$ using the context vectors.

To prove this we consider two models. The original model is designed to solve task 1. The PSP model is combining models for several tasks. Consider the original model as a function of its parameters $w$ and denote it as $f(w)$. Throughout this section we assume $w$ to be a vector.

Note that for every $w$, the function $f(w)$ defines a mapping from inputs to outputs. The PSP model, when used for task 1, can also be defined as $F(W)$, where $W$ is a superposition of all weights.

We define a superposition function $\varphi$, combining weights $w$ with any other set of parameters, $\tilde{w}$.

$$W = \varphi(w, \tilde{w})$$

We also define an read-out function $\rho$ which extracts $w$ from $W$ possibly with some error $e$:

$$\rho(W) = w + e.$$

The error $e$ must have such properties that the two models $f(w)$ and $f(w+e)$ produce similar outputs on the data and have approximately equal gradients $\nabla_w f(w)$ and $\nabla_w f(w + e)$ on the data.

When the PSP model is used for task 1, the following holds:

$$F(W) = f(\rho(W)) = f(w + e)$$

Our goal now is to find conditions of superposition and read-out functions, such that for any input/output data the gradient of $f$ with respect to $w$ is equal (or nearly equal) to the gradient of $F$ with respect to $W$, transformed back to the $w$ space. Since for the data the functions $f(w)$ and $f(w + e)$ are assumed to be nearly equal together with their gradients, we can omit the error term $e$.

The gradient updates the weights $W$ are

$$\delta W = \nabla_W F$$

$$= \left(\frac{\partial \rho}{\partial W}\right)^T \nabla_\rho f(\rho(W))$$

$$= \left(\frac{\partial \rho}{\partial W}\right)^T \nabla_w f(w)$$

Now $\delta w$ corresponding to this $\delta W$ can be computed using linear approximate of $\rho(W + \delta W)$:

$$\delta w = \rho(W + \delta W) - \rho(W) \approx \frac{\partial \rho}{\partial W} \delta W,$$

and hence

$$\delta w = \left(\frac{\partial \rho}{\partial W}\right) \left(\frac{\partial \rho}{\partial W}\right)^T \nabla_w f(w)$$

Thus in order for $\delta w$, which is here computed using the gradient of $F$, to be equal to the one computed using the gradient of $f$ it is necessary and sufficient that

$$\left(\frac{\partial \rho}{\partial W}\right) \left(\frac{\partial \rho}{\partial W}\right)^T = I \tag{10}$$

**Real-valued network with binary context vectors**  Assume a weights vector $w$ and a binary context vector $b \in \{-1, 1\}^M$. We define a superposition function

$$W = \varphi(w, \tilde{w}) = w \odot b + \tilde{w}.$$

Since $b = b^{-1}$ for binary vectors, the read-out function can be defined as:

$$\rho(W) = W \odot b$$
$$= w + \tilde{w} \odot b$$
$$= w + e$$

In propositions 1 and 2 we have previously shown that in case of binary vectors the error $e$ has a small contribution to the inner product. What remains to show is that the condition 10 is satisfied.

Note that

$$\frac{\partial \rho}{\partial W} = \text{diag}(b).$$

Since $b_i b_i = 1$ for every element $i$, the matrix $\frac{\partial \rho}{\partial W}$ is orthogonal and hence condition 10 is satisfied.

$$\left(\frac{\partial \rho}{\partial W}\right) \left(\frac{\partial \rho}{\partial W}\right)^T = \text{diag}(b)\text{diag}(b)^T$$
$$= I$$

**Complex-valued network with complex context vectors.** The proof for the complex context vectors is very similar to that for the binary. Let the context vector $c \in \mathbb{C}^M$, s.t. $|c_i| = 1$. It is convenient to use the notation of linear algebra over the complex field. One should note that nearly all linear algebraic expressions remain the same, except the transposition operator $^T$ should be replaced with the Hermitian conjugate $^\dagger$ which is the combination of transposition and changing the sign of the imaginary part.

We define the superposition operation as

$$W = \varphi(w, \tilde{w}) = w \odot c + \tilde{w}.$$

When $|c_i| = 1$, $c_i^{-1} = c_i^*$ where $^*$ is the element-wise conjugate operator (change of sign of the imaginary part). The read-out function can be defined as:

$$\begin{aligned} \rho(W) &= W \odot c^* \\ &= w + \tilde{w} \odot c^* \\ &= w + e \end{aligned}$$

The necessary condition 10 transforms into

$$\left(\frac{\partial \rho}{\partial W}\right)\left(\frac{\partial \rho}{\partial W}\right)^\dagger = I$$

where $I$ is a complex identity matrix, whose real parts form an identity matrix and all imaginary parts are zero. This condition is satisfied for the chosen complex vector because

$$\left(\frac{\partial \rho}{\partial W}\right)\left(\frac{\partial \rho}{\partial W}\right)^\dagger = \mathrm{diag}(c)\mathrm{diag}(c^*) = I$$

**Real-valued network with real-valued rotational context matrices** The proof is again very similar to the previous cases. The superposition operation is defined as

$$W = Cw + \tilde{w},$$

where $C$ is a rotational matrix.

The read-out function is

$$\rho(W) = C^{-1}W = C^T W$$

where $C^T$ is the transposed of $C$. Here we use the fact that for rotation matrices $CC^T = C^T C = I$.

The condition 10 becomes

$$\left(\frac{\partial \rho}{\partial W}\right)\left(\frac{\partial \rho}{\partial W}\right)^T = C^T C = I,$$

and hence is satisfied. $\qquad\square$

## C  Geometry of Rotations

For each type of superposition, Supplementary Figure 1 left provides the geometry of the rotations which can be applied to parameters $w$. This illustrates the topology of the embedding space of superimposed models.

## D  From superposition to composition

While a context is an operator on parameter vectors $w$, the context itself can also be operated on. Analogous to the notion of a group in abstract algebra, new contexts can be constructed from a composition of existing contexts under a defined operation. For example, the context vectors in complex superposition form a Lie group under complex multiplication. This enables parameters to be stored and recovered from a composition of contexts:

$$c^{a+b} = c^a \odot c^b \tag{11}$$

By creating functions $c(k)$ over the superposition dimension $k \in S$, we can generate new context vectors in a variety of ways. To introduce this idea, we describe two basic compositions.

Supplementary Figure 1: The topology of all context operators acting on a vector $w$, e.g. $\mathbb{T}^M \cdot w = \{c \odot w : c \in \mathbb{T}^M\}$. **A** binary operates on a lattice **B** complex operates on a torus **C** rotational operates on a sphere.

Supplementary Figure 2: Accuracy on rotating MNIST

**Mixture of contexts**    The continuity of the phase $\phi$ in complex superposition makes it possible to create mixtures of contexts to generate a smoother transition from one context to the next. One basic mixture is an average window over the previous, current and next context:

$$c(k) = e^{i\frac{\phi(k-1)+\phi(k)+\phi(k+1)}{3}} \tag{12}$$

The smooth transitions reduces the orthogonality between neighboring context vectors. Parameters with neighboring contexts can 'share' information during learning which is useful for transfer-learning settings and continual learning settings where the domain shift is smooth.

# E   Additional Results

Supplementary Figure 3: Comparing the accuracy of various methods for PSP on the first task of the permutingMNIST challenge over training steps on networks with 2000 units. pspRotation is left out because it is impractical for most applications because its memory and computation footprint is comparable with storing independent networks.