[Reviews · NeurIPS 2019]

Reviewer 1



-------------------------------------------------------------------------------------------------------- Post-rebuttal update: I have now read the rebuttal and the other reviews. I appreciate that the authors re-implemented the CIFAR benchmark I had requested. However, I'm still unconvinced of the significance or the originality of the proposed approach. For me, two fundamental issues remain: 1) The proposed approach is conceptually very similar to the masking proposed in Masse et al. (2018) that I mentioned in my review. The only difference is essentially masking with a {1,-1} vector vs. masking with a {0,1} vector. For sufficiently sparse masks (as used in Masse et al.), the latter approach will also produce largely non-overlapping feature subsets for different tasks, so I don't see this as a huge difference. The authors respond that their approach is more general. That may be so, but is this enough of a conceptual advance? We don't even know how well the proposed approach works compared to the {0,1} masking, because the authors have chosen to largely ignore the prior literature in their evaluations. Relatedly, Masse et al. also show that one has to combine masking with a synaptic stabilization mechanism such as EWC in order to get the best results. Is this also the case in the current paper as well? Again, we don't know, because the evaluations in the current paper are not thorough enough. 2) The authors are essentially claiming that doing something random (i.e. random masking) works better than doing something intelligent, like considering the importance of different parameters for prior tasks as in EWC, and I feel very uneasy about this claim. I feel like there's an important catch that isn't currently being made clear in the paper. In their rebuttal, the authors say that "methods like EWC, GEM constraint the amount of change in parameters for a new task in a manner that ensures performance on past tasks is maintained. With increase in tasks, constraints accumulate, making it impossible to add new tasks once optimization slows. Our method, however, shows a different property – its much more flexible in the sense that it forgets the older tasks when network capacity is reached in order to accommodate newer tasks." So maybe this is the catch that needs to be made really clear (but currently isn't): for a large enough number of tasks, the proposed method will fail much more rapidly in prior tasks than methods like EWC or GEM, but again we aren't really sure if and when this will happen (or relatedly, whether a combination of masking and stabilization as in Masse et al. might strike the best balance by guaranteeing both learning and retention), because again the experiments in the paper aren't thorough enough. So, in conclusion, given the two lingering fundamental issues I have, I have decided to keep my score as it is. -------------------------------------------------------------------------------------------------------- - In general, the comparison with earlier methods is not thorough at all in this paper. Several important earlier works have not been discussed: Schwarz et al. (2018) Compress & Progress: A scalable framework for continual learning. ICML Lopez-Paz & Ranzato (2017) Gradient episodic memory for continual learning. NIPS - The proposed approach is actually quite similar to the one proposed in the following paper (this paper has been on arxiv since Feb. 2018): Masse et al. (2018) Alleviating catastrophic forgetting using context-dependent gating and synaptic stabilization. PNAS Yet, this paper is also not discussed, cited, or compared against. - Conceptually, the proposed method is also quite similar to the "sparsely-gated mixture-of-experts" model (Shazeer et al, 2018, ICLR). The difference is that the gating is learned in that paper (which may actually be more sensible than using a fixed, random gating) and it's applied on a per-example basis, instead of a per-task basis. - The CIFAR-100 experiment is not implemented in a standard way (please see Lopez-Paz & Ranzato, 2017, cited above). Earlier papers use a larger number of tasks (20 tasks in Lopez-Paz & Ranzato, 2017). Please implement this experiment consistently with earlier papers, and compare your numbers with earlier methods. - There is, similarly, a question about the scalability of the proposed method to more challenging tasks. For example, the EWC paper (and the C&P paper above) had Atari benchmarks (including a setup where the task identity was automatically inferred instead of being manually set). I understand that the main advantage of the proposed method is its simplicity, but if it doesn't scale up to more challenging tasks, its significance will be limited. So, I would encourage the authors to test their method on Atari benchmarks and compare with earlier methods. - Because of the random nature of the context variables, it seems to me that the proposed method does not allow transfer learning, which is not ideal. I would encourage the authors to think about extending their method to enable transfer learning while retaining its simplicity as much as possible. For example, at the beginning of each new task, one can fix the weights of the network, and train the context variables first (to encourage the network to utilize the already learned features), then fix the context variables and fine-tune the network weights etc. - In several places, the authors talk about the low intrinsic dimensionality of natural signals, which is necessary to make the context-gated inputs for different tasks distinct from each other. But since the authors are applying the context gating at every layer of a network, the low intrinsic dimensionality of the raw inputs is not enough. The input to each layer should be low-dimensional. This is clearly not the case for higher layers of a conv-net, for example (or at least, they're nowhere near as redundant as the raw input). The authors should discuss this important caveat. More minor: - There are several typos: line 34 (" ... training a neural networks in..."); line 44 (should be "separate tasks"); line 135 (should be "evaluate the performance of the..."); line 216 (should be "it is worse in performance than..."); line 238 (should be "change"). - The description of the proposed method can be improved. For example, the context variables are fixed and random; they are not trainable. But, this important detail is somehow never explicitly stated in describing the method. Worse, the authors keep calling these "parameters" as if they are like the trainable parameters of the network. I'm sure a lot of readers will be very confused by this. - Line 209: the authors say that it is beyond the scope of the paper to propose a solution to the problem of automatic inference of task identity, but I personally think this would make the paper more compelling. For example, in the rotation example, the circular nature of the transformations is assumed to be known already, which is clearly unrealistic in the general case. So I would again encourage the authors to think about this issue. - In Figure 5 (right), what is pspLocalMix? This is not defined anywhere in the text or the caption (pspFastLocalMix is defined). - Please make the figure fonts bigger; currently they are very hard to read without zooming in.

Reviewer 2



I have a few comments which I hope Authors can address to make this paper stronger and more complete: 1- Related work is minimal. Although there is a short paragraph compare your results with previous work, but it is not enough. The reader should get more insight about previous approaches and a high-level understanding of why previous approaches are different than yours. Please expand this section. 2- There are some confusions and inconsistencies in experimental results, section 4.1.3, paragraph "choosing context parameters": 2.1- I see in Figure 5-right that you present results for pspLocalMix, but I couldn't find what approach it refers to. 2.2- The text says: "... Figure 5 right shows that while pspFast is better than standard model ..." but standardmod results are not in Figure5-right. It is a combination of Figure 4-b and Figure 5, right? 2.3- Does anywhere in the text talks about Figure5-left? 3- I think how many models can be stored in superposition with each other is a very interesting question and shouldn't be left out in this paper. I know Authors mentioned this in Future work and Discussion, but this is a fundamental question that directly related to the effectiveness of their proposed solution and at least some preliminary empirical results are required to be included in this paper. I understand Authors have realized degradation in performance while keeping 1000 models superposition-ed with each other, but it is very interesting to get more insight about the limitation of their approach in terms of the number of models stored together.

Reviewer 3



# Strengths: - The paper copes with an interesting problem. Both model compression and lifelong learning are active domains of research and proposing an approach which somehow bridges the two problems and proposes a dual view on these problems is great. - The exposition of the paper is very clear. The paper reads nicely and the problem is very well motivated. - The proposed algorithm is very simple and therefore easy to understand. Its simplicity should definitely be considered as a feature. - The approach is evaluated in multiple experiments, some of them on "controlled" data, and one experiment on a real model (ResNet-18) on real data (CIFAR-10 CIFAR-100). The experiments are quite convincing. - The proposed model compares favorably with previous work (See Fig. 3). # Weaknesses - When reading papers about catastrophic forgetting, I can't help myself from asking for actual practical applications. All experiments in this paper are on datasets that have been actually made up from standard benchmarks, by concatenating manipulated samples. There are probably real tasks with available data for which it would make sense to use such a model. I would like the authors to comment on that. - What lessons learnt from this work could influence or guide research in model compression? Overall, I enjoyed reading this paper who's motivations are clear and execution is very good. The tackled problem is interesting and the experiments convincing. Please note however that I am not an expert in lifelong learning, and may have missed important details or relevant work that could change my opinion. Therefore, I await the author's response, the other reviews and discussion to make my final decision.

[Author Response · NeurIPS 2019]

We thank the reviewers for their feedback. R2 and R4 recommend accepting our paper based on a novel, mathematically well motivated approach and convincing results. R2 says, *" .. a fundamentally different way of diminishing catastrophic forgetting.. proposed solution is backed by solid mathematical framework .. "*; R4 says, *"..an approach which somehow bridges the two problems and proposes a dual view on these problems is great.. experiments are quite convincing.."* R1 raised concerns regarding comparison to prior work and specific contributions of our work, which we address first.

**R1: "Re-implement CIFAR-100 experiments .. compare with EWC, GEM"**
A: We initially followed the experimental protocol of Zenke et. al. 2017 for the CIFAR experiments. We re-implemented the variant in Lopez-Paz 2017. Our method achieves 65.2% average accuracy across 5 runs for ResNet18, significantly outperforming EWC (49.8% accuracy), iCARL (Rebuffi et al., 54.6%) and comparable to GEM result of 67.8%. Note that the results of prior work were computed in the GEM paper using a non-standard ResNet architecture and required a separate output layer for each task. In contrast, we use the standard ResNet18 and only require a single output layer.

**R1: " .. several important earlier works have not been discussed .."**
A: We acknowledge the brevity of our discussion of past work in our paper and promise to significantly expand this discussion in the next draft. The key differences/benefits of our work are:

• We require significantly less number of additional parameters to add a new task. For instance, is a fully connected layer has MxN parameters, we only require M additional parameters (binaryPSP) for each new task. In contrast, EWC stores the diagonal Fisher Matrix of size MxN parameters for each task.

• Lopez-Paz et al. necessarily need to store data from *all* the previous tasks. Our method has no such requirements.

• Masse et al.'s idea of gating is conceptually different from our method. Masse et al., gate by zeroing out a subset of features (i.e. a bitwise multiplication with a vector such as 1, 0, 1, 0), whereas we use the rotation of features to nearly orthogonal vectors. E.g. in binary superposition, a context vector might be: -1, 1, 1, -1. However, we can choose the context to be ternary (e.g. -1, 0, 1, 0), where a few elements are set to 0. The gating method of Masse et al. is therfore a special case of our framework. We will include our results showing that ternary superposition is superior to gating by simply zeroing out features. Additionally, we have provided theoretically motivated guidelines for choosing these basis/context vectors and the effect of these choices on the amount of catastrophic forgetting.

• Methods like EWC, GEM constrain the amount of change in parameters for a new task in a manner that ensures performance on past tasks is maintained. With increase in tasks, constraints accumulate, making it impossible to add new tasks once optimization slows. Our method, however, shows a different property – its much more flexible in the sense that it forgets the older tasks when network capacity is reached in order to accommodate newer tasks.

**R1: " ..context variables are fixed and random; not trainable .. one can train the context variables.."**
A: Great question and suggestion. We will clarify this further in the text. We chose random context variables because randomness is enough to guarantee (near) orthogonality of vectors in high-dimensional vector spaces. However, our framework is fully differentiable and it is possible to learn the context variables as you suggested. In fact, we have been working in this direction and the initial results are very promising.

**R1: " .. automatic inference of task identity .."**
A: This question is of great interest and we are working on it, but is complementary to our main contribution of demonstrating that parameter superposition is a powerful tool to combat catastrophic forgetting.

**R1: "If possible, implement the Atari benchmarks and compare the results with EWC."**
A: Unfortunately, we could not train on ATARI due to time constraints – but will include in the final version.

**R1, R2: A few typos and inconsistencies in present in the results.**
A: Sorry for the confusion. We promise to revise the text to improve clarity and remove typos and any inconsistencies.

**R2: " ..how many models can be stored in superposition is a very interesting question .."**
A: Yes, indeed. On permuting MNIST, we tested networks with 2000, 1000, 500, 250 units and each storing 1600, 800, 400, 200, 100 different tasks. Performance degrades gracefully with number of tasks: 2000 unit network has an average accuracy of (94.06%, 100 tasks); (92.35%, 200 tasks); (85.21%, 400 tasks); (65.15%, 800 tasks). For a 250 unit network: (83.68%, 100 tasks); (72.46%, 200 tasks); (53.46%, 400 tasks); (35.65%, 800 tasks). We will include these results in the final version of the paper.

**R3: "..real tasks with available data for which it would make sense to use such a model..authors to comment.."**
A: Continual learning methods are very useful in resource constrained online -learning settings, where it is not possible to store past data to form batches or in situations where data statistics are continuously changing. An example is mini-drone performing remote surveillance as lighting changes from day to night (analogous to rotating MNIST).

**R3: "..how this work could influence or guide research in model compression?"**
A: Model compression at training time is an unsolved problem. We present one way to use excess capacity in networks that solve multiple tasks during training time – we hope this instigates more interest in online compression of models.

[Meta-Review · NeurIPS 2019]

The reviewers did not come to a consensus on the merits of this paper. In my opinion, the approach presented is novel and interesting; but I share R1's concerns about the discussion of earlier work and lack of empirical comparisons. I am recommending the paper be accepted, but the authors should include a discussion of these prior works, and comparisons to relevant methods where appropriate.